# Mechanistic Integration of Network Pharmacology and In Vivo Validation: TFRD Combat Osteoporosis via PI3K/AKT Pathway Activation

**DOI:** 10.3390/ijms26083650

**Published:** 2025-04-12

**Authors:** Chang Tan, Shibo Cong, Yanming Xie, Yingjie Zhi

**Affiliations:** 1Institute of Basic Research in Clinical Medicine, China Academy of Chinese Medical Sciences, Beijing 100700, China; tc521@outlook.com; 2Institute of Basic Theory for Chinese Medicine, China Academy of Chinese Medical Sciences, Beijing 100700, China; shiboc@bucm.edu.cn

**Keywords:** total flavonoids of rhizoma drynariae, TFRD, ovariectomized osteoporosis model, PI3K/AKT signaling pathway, network pharmacology, bone metabolism

## Abstract

In the context of osteoporosis closely linked to bone metabolism imbalance caused by estrogen deficiency, total flavonoids of Rhizoma Drynariae (TFRD) exhibit potential anti-osteoporotic activity, yet their multicomponent synergistic mechanism and association with the PI3K/AKT signaling pathway remain unclear. This study aimed to systematically elucidate the molecular mechanisms by which TFRD regulate bone metabolism and improve osteoporosis in ovariectomized (OVX) rats through the PI3K/AKT pathway, integrating network pharmacological predictions with animal experimental validation. Methods involved identifying TFRD’s active components using UPLC/MS-MS, predicting targets with SwissTargetPrediction, constructing a “component-target-disease” network, and performing GO/KEGG enrichment analysis with MetaScape (v3.5). In vivo experiments established an OVX rat model, randomized into sham, OVX, low-/high-dose TFRD, and sim groups, assessing bone mineral density (BMD) and mandibular Micro-CT parameters after 12 weeks. Western blot analyzed PI3K, p-AKT1, and related protein expressions. Results showed the high-dose TFRD group significantly increased BMD, improved trabecular bone quantity and structure, and upregulated PI3K, p-PI3K, and p-AKT1 protein expressions compared to the OVX group. Molecular docking confirmed stable binding energy between core components and PI3K/AKT targets. TFRD may ameliorate estrogen deficiency-induced osteoporosis by activating the PI3K/AKT signaling pathway, inhibiting bone resorption, and promoting osteogenic differentiation, providing pharmacological evidence for multitarget treatment of osteoporosis with traditional Chinese medicine.

## 1. Introduction

Osteoporosis (OP) is a skeletal disease that reduces bone mass and damages the microstructure of bone tissue, increasing fragility and fracture risk [1]. It is a global public health issue, seriously affecting the quality of life of middle-aged and elderly people [2]. The pathogenesis of this disease is complex, involving factors like genetics, hormones, nutrition, and lifestyle. The main cause of osteoporosis in postmenopausal women is estrogen deficiency [3]. Postmenopausal women with systemic osteoporosis also experience jaw-bone loss, making their teeth more likely to loosen and fall out. Osteoporosis affects the density and structure of alveolar bone, leading to resorption and damage. Changes in alveolar bone density and structure can compromise tooth stability, causing them to loosen, shift, or even fall out [4]. Patients with osteoporosis are at greater risk of losing teeth, which affects oral health and can impair chewing and nutrient intake [5]. Alveolar bone loss is closely linked to systemic osteoporosis [6]. Current treatment strategies primarily aim to inhibit bone resorption, enhance bone formation, or address underlying metabolic deficiencies. First-line pharmacological agents include antiresorptive drugs, such as bisphosphonates (e.g., alendronate and zoledronate), which suppress osteoclast activity by inhibiting farnesyl pyrophosphate synthase in the mevalonate pathway [7,8]. Denosumab, a fully human monoclonal antibody targeting RANKL, has shown superior efficacy in reducing vertebral fracture risk compared to placebo in phase III trials [9]. For patients with severe osteoporosis or those who fail antiresorptive therapy, anabolic agents like teriparatide (recombinant parathyroid hormone 1–34) and romosozumab (a sclerostin inhibitor) are recommended. Teriparatide stimulates osteoblast differentiation via PTH receptor signaling, while romosozumab transiently activates Wnt/β-catenin pathways to enhance bone formation [9]. Adjunctive therapies include calcium and vitamin D supplementation, which remain essential for maintaining skeletal health, particularly in patients with nutritional deficiencies [10]. However, emerging evidence highlights limitations in current regimens, such as long-term safety concerns with bisphosphonates (e.g., atypical femoral fractures) and high costs of biologics [11,12]. This underscores the urgent need for novel therapeutic targets and personalized intervention strategies.

Drynaria fortunei, a traditional Chinese medicine, is commonly used to treat fractures, osteoporosis, and other bone diseases [13]. Its main active component, total flavonoids, has gained significant attention for its strong bone-protective effects [14]. The total flavonoids from Rhizoma Drynariae (TFRD) can promote bone formation and inhibit bone resorption [15]. Research indicates that these flavonoids influence key factors like the receptor activator of nuclear factor-κB ligand (RANKL) and estradiol (E2), regulating bone metabolism [16]. They enhance osteoblast activity, support bone matrix formation, and simultaneously reduce osteoclast activity to decrease bone resorption. This helps maintain a balance between bone formation and resorption, improving overall bone quality [17]. While TFRD show promise for treating OP, their specific mechanism of action remains unclear. The PI3K(phosphatidylinositol-3kinase)/AKT (protein kinase B) signaling pathway is a key intracellular signal transduction route that regulates various cellular functions, including osteoblast and osteoclast activities, thus affecting the balance between bone formation and resorption [18]. Studies suggest that this pathway plays a vital role in the development of OP [19]. Exploring whether TFRD have therapeutic effects on osteoporosis through the PI3K/AKT pathway is important for understanding its pharmacological mechanism and improving treatment strategies.

This study systematically investigates how TFRD treat OP in OVX rats through the PI3K/AKT pathway, using network pharmacology and in vivo experiments. It is significant because it reveals the specific mechanism of action for these flavonoids, providing a theoretical basis and experimental evidence for modern research and clinical applications of TCM. Additionally, by examining the role of the PI3K/AKT pathway in treating OP, it identifies potential targets for new anti-OP drugs and supports the theory of “treating different diseases with the same therapy” in TCM.

## 2. Results

### 2.1. Results of Network Pharmacology Analysis

#### 2.1.1. TFRD Component Target Prediction

In our previous study, we performed UPLC-MS/MS analysis on TFRD and identified 23 active components (Appendix A). The specific parameters are in the Appendix A. Using SwissTargetPrediction, we searched for these compounds and predicted 556 disease targets based on their structures. By combining DISGENET, GENECARDS, OMIM, CTD, and TTD databases with “osteoporosis” as a keyword, we found 5718 OP-related targets after removing duplicates. We used R 4.3.1 software to create a Venn diagram (Figure 1A) and identified 307 common targets for TFRD in treating OP. These common targets may be key for TFRD’s effectiveness against OP and provide valuable insights for future experimental verification.

#### 2.1.2. PPI Network Analysis of TFRD’s Common Action Targets in Treating OP

Based on the key targets for TFRD in treating OP, a PPI network was created using the STRING database, followed by topological analysis. The network was visualized and analyzed with Cytoscape 3.7.2. Results showed that core proteins of the PI3K/AKT and JAK/STAT3 pathways formed distinct modules within the network, closely interacting with TFRD’s predicted targets (Figure 1B). Comparing these targets to known proteins in the PI3K/AKT pathway revealed many had direct or indirect interactions with its core proteins, including PI3K, AKT1, and TP53—all crucial players in this pathway. This supports the prediction that TFRD may exert their therapeutic effects on OP through the PI3K/AKT pathway.

#### 2.1.3. Construction of the “TFRD Component-Targets” Network

We created a network diagram of TFRD components and OP targets, revealing significant interactions between them. TFRD components can directly bind to PI3K, AKT1, and Tp53, forming stable complexes (Figure 1C). They also affect other signaling pathways related to OP targets indirectly. For example, TFRD components regulate the JAK/STAT3 inflammatory pathway, reducing inflammation.

#### 2.1.4. Pathway Enrichment Analysis

In this study, we performed pathway analysis on intersecting genes using the KEGG database. The results revealed significant enrichment in multiple pathways, particularly the PI3K/AKT pathway, which plays a crucial role in the potential mechanism of TFRD for treating OP (Figure 2A). By comparing different pathways’ enrichment levels, we found that the PI3K/AKT pathway is closely linked to genes like PI3K, AKT1, TP53, and JAK. Additionally, other bone metabolism-related pathways such as the EGFR and Endocrine resistance pathways were also involved. These may work together with the PI3K/AKT pathway to regulate bone metabolism processes. This finding offers valuable insights into understanding how PI3K/AKT-related genes function in biological processes and suggests new directions for drug development and disease treatment.

We also performed a GO analysis on the intersection genes of TFRD and OP to understand their roles in biological processes. The results (Figure 2B) indicated that in the molecular function category, these genes were significantly enriched in “membrane raft” and “membrane microdomain”, suggesting their involvement in cell membrane metabolism. In the biological process category, they were notably enriched in a “cellular response to chemical stress” and “response to oxidative stress”, highlighting their role in regulating chemical and oxidative stress. These findings offer valuable insights into the functions of these intersection genes.

#### 2.1.5. “TFRD Component-Target-Pathway” Network

In network pharmacology research, it is essential to explain how TFRD components interact with specific OP targets and influence related biological pathways. This study created a diagram illustrating the relationships between TFRD components, OP targets, and pathways. The diagram shows that TFRD components significantly interact with multiple OP targets (Figure 2C). Notably, they can directly bind to PI3K/AKT pathway-related targets, forming stable complexes that control cell proliferation and exert anti-OP effects. These findings offer new insights into the anti-OP mechanism of TFRD components and provide valuable references for future drug development.

### 2.2. In Vivo Experimental Results

#### 2.2.1. Bone Morphometry and BMD Analysis

After the administration was completed, we measured the BMD of the mandible in rats and conducted histological section observations. Through H&E staining (Figure 3A), we observed the morphological changes in the mandibular alveolar bone tissue in rats. The results showed that in the CON (Control) group and the SOG (Sham Operation Group), the rats’ mandibular alveolar bone pulp had rich blood vessels and nerves, and there were many pulp mesenchymal cells, odontoblasts, and stellate pulp cells, which were evenly distributed, and the tissue structure was normal. However, in the OVX rats, the trabecular bone structure was sparse, the bone cells were arranged disorderly, the bone mass was significantly reduced, the pulp cells were significantly decreased, the collagen fibers increased, showing degenerative changes, and only a few pulp mesenchymal cells and blood vessels were generated. In the TFRD treatment group and the positive drug group, the pulp cells in the mandibular alveolar bone of rats increased and were arranged orderly, and a large number of new blood vessels were formed, the collagen fibers were significantly reduced, the trabecular bone structure was improved compared with the OVX group, the bone cells were relatively orderly arranged, and the bone mass increased.

#### 2.2.2. The Impact of TFRD on the BMD in Ovariectomized (OP) Rats

Compared to the SOG (1.09 ± 0.07), rats in the OVX group (0.66 ± 0.1) showed a significant decrease in bone density (*p* < 0.01), confirming that ovariectomy effectively induced OP. The mandibular bone density of rats in both the TFRD-H (high-dose TFRD) (0.81 ± 0.12) and Sim(Positive drug) groups (0.79 ± 0.06) was significantly higher than that of the OVX group (0.66 ± 0.1) (*p* < 0.05), with improved trabecular number, density, and distribution as well (Figure 3C). This indicates that TFRD can enhance bone density in OVX rats. Although there was no significant difference between the TFRD-L (low-dose TFRD) (0.78 ± 0.04) and OVX groups, an upward trend was observed, suggesting potential benefits for bone quality and morphology. Overall, these findings highlight the preventive and therapeutic effects of TFRD on postmenopausal OP.

#### 2.2.3. The Impact of TFRD on the BV/TV Ratio in Ovariectomized (OP) Rats

Compared to the SOG (24 weeks) (86.26 ± 6.63), the BV/TV values of the mandible were significantly lower in both the OVX group (50.64 ± 12.31) and the Sim group (72.72 ± 6.49), as well as in the TFRD group (TFRD-L: 57.78 ± 10.99) (TFRD-H: 76.91 ± 9.66) (*p* < 0.01) (Figure 3D). However, compared to the OVX group (50.64 ± 12.31), these values increased significantly in both the Sim and TFRD groups (*p* < 0.05).

#### 2.2.4. The Impact of TFRD on the Tb.Th Ratio in Ovariectomized (OP) Rats

Compared to the SOG (0.23 ± 0.04) at 24 weeks, the Tb.Th value of the mandible in the OVX group (0.13 ± 0.03) was significantly lower (*p* < 0.01) (Figure 3E). In contrast, both the Sim (0.18 ± 0.02) and TFRD groups (TFRD-L: 0.16 ± 0.02) (TFRD-H: 0.21 ± 0.02) showed a significant increase in Tb.Th values compared to the OVX group (*p* < 0.01, *p* < 0.05), but these values remained significantly lower than those of the SOG (*p* < 0.01).

#### 2.2.5. The Impact of TFRD on the Tb.N Ratio in Ovariectomized (OP) Rats

There was no significant difference in the Tb.N of lumbar vertebrae among the rat groups after treatment (*p* > 0.05) (Figure 3F). (SOG: 3.82 ± 0.59; OVX: 3.27 ± 0.25; TFRD-L: 3.31 ± 0.36; TFRD-H: 3.4 ± 0.49; Sim: 3.37 ± 0.29).

#### 2.2.6. The Impact of TFRD on the Tb.Sp Ratio in Ovariectomized (OP) Rats

Compared to the SOG (24 weeks) (0.1 ± 0.04), the Tb.Sp values in the mandible were significantly higher in the OVX group (24 weeks) (0.31 ± 0.05), Sim (0.24 ± 0.03), and TFRD group (TFRD-L: 0.25 ± 0.04) (TFRD-H: 0.2 ± 0.03) (*p* < 0.01) (Figure 3G). Additionally, compared to the OVX group (24 weeks), these values were significantly lower in both the Sim and TFRD groups (*p* < 0.01, *p* < 0.05).

#### 2.2.7. Detection of Protein Levels in the JAK/STAT3 and PI3K/AKT Pathways

We used Western blot to measure the expression levels of key proteins in the JAK/STAT3 and PI3K/AKT pathways (like PI3K, AKT1, and TP53) in rat mandibular bone tissue. The results indicated that, compared to the SOG, the expression levels of JAK (p-JAK) and STAT3 (p-STAT3) proteins were significantly lower in the OVX group (*p* < 0.01) (Figure 4A,B). Additionally, after 24 weeks, the TFRD group showed a significant increase in these protein levels compared to the OVX group (*p* < 0.05, *p* < 0.01).

The TFRD group showed significantly higher levels of PI3K, p-PI3K, and p-AKT1 compared to the OVX group (*p* < 0.01) (Figure 4C,D), suggesting that TFRD may have therapeutic effects by regulating key proteins in the PI3K-AKT pathway. Further investigation revealed that AKT1 interacts with TP53 via p-MDM2, both being downstream of AKT1. Thus, we also measured protein levels of p-MDM2 and TP53. Analysis showed that in the OVX group (Figure 4E,F), levels of PI3K, p-PI3K, p-AKT1, and p-MDM2 were significantly lower than in the SOG (*p* < 0.01), indicating inhibition of the PI3K/AKT pathway in OVX rats. However, these protein levels were significantly higher in the TFRD group compared to the OVX group (*p* < 0.01), while TP53 exhibited an opposite trend. This indicates that TFRD can up-regulate key proteins in this pathway and may exert therapeutic effects through it.

#### 2.2.8. Molecular Docking Verification

We conducted molecular docking verification to confirm the interaction between key active components in TFRD and proteins in the PI3K/AKT pathway. Based on our network pharmacology analysis results, we selected the proteins most related to these active components as targets for docking simulations using AutoDockVINA.

The docking results (Table 1) indicated that the binding energy of Luteolin with proteins such as PI3K and AKT1 was relatively low, suggesting a strong interaction between them. This further confirmed that Luteolin might regulate the PI3K/AKT pathway by modulating the activity of PI3K (Figure 5).

## 3. Discussion

This study systematically analyzes the chemical components of TFRD and their potential target proteins using network pharmacology methods, thereby constructing a drug-target-pathway network. Furthermore, through enrichment analysis, we identify several significant signaling pathways and biological processes associated with the components of TFRD, which may represent key mechanisms by which these components exert their therapeutic effects (Figure 1). This analysis not only identifies the presence of multiple active components in TFRD but also elucidates the central role of the PI3K/AKT signaling pathway within this context (Figure 2A), thereby demonstrating the advantages inherent in TCM’s coordinated intervention across multiple targets and pathways [20]. The results of the network pharmacology analysis indicate that multiple components of TFRD can interact, either directly or indirectly, with key proteins in the PI3K/AKT signaling pathway, thereby modulating its activity. This finding provides a theoretical foundation for subsequent experimental validation and delineates specific directions for future research.

To validate the predictive outcomes of network pharmacology, we conducted in vivo experiments utilizing OVX osteoporotic rats as our experimental model. We administered TFRD via gavage and assessed their impact on bone mass and microstructural integrity (Figure 3). Following the conclusion of the administration period, we employed Micro-CT technology to evaluate the BMD of the mandible in these rats. Our findings indicated that the BMD in the treatment group was significantly greater than that observed in the OVX group (*p* < 0.05) (Figure 3C). Furthermore, we performed histological examinations of the mandible. The findings indicated that in the drug administration group, the number of odontoblasts in the rats increased and exhibited an orderly arrangement, accompanied by a significant presence of newly formed blood vessels. Notably, collagen fibers were markedly reduced, and there was an improvement in trabecular bone structure compared to the OVX group (Figure 3A). Bone cells were relatively well-organized, and bone mass showed an increase. These results preliminarily confirm that TFRD can significantly enhance both bone quality and morphology in OVX rats.

To further elucidate its mechanism of action, we employed Western blot technology to assess the expression levels of key proteins in the PI3K/AKT signaling pathway within rat mandibular bone tissues. Our findings demonstrated that TFRD significantly enhanced the expression of PI3K, p-PI3K, and p-AKT1 (*p* < 0.01), indicating activation of the PI3K/AKT pathway (Figure 4). These experimental results were highly consistent with those obtained from network pharmacology analysis, thereby providing additional validation that TFRD may enhance bone metabolism processes in OVX rats by modulating the PI3K/AKT pathway and associated signaling cascades. Molecular docking analyses further substantiated the robust binding affinity of the core components to pivotal proteins within the PI3K/AKT signaling pathway, including PI3K and AKT1. In combination with the literature reports and existing experimental data, we found that Luteolin exhibited regulatory effects on the PI3K/AKT pathway in various cell models [21,22]. These results further verified our molecular docking results and indicated that components such as Luteolin in TFRD might exert therapeutic effects on OVX OP through this pathway. This provides a theoretical foundation for subsequent in vivo experiments. Moreover, this analytical approach not only elucidated potential target sites of TFRD but also established a methodological framework for investigating the mechanisms underlying other TCM formulations.

The PI3K/AKT signaling pathway is a critical intracellular signal transduction mechanism that plays an essential role in regulating cell proliferation, differentiation, apoptosis, and metabolism [23]. In the context of osteoporosis, dysregulation of the PI3K/AKT pathway may result in functional impairments of bone cells, thereby adversely affecting the processes involved in bone metabolism [18]. Modulating the PI3K/AKT pathway could be an effective way to treat osteoporosis. This mechanism explains how TFRD improve osteoporosis and support its use in treatment. We also investigated potential interactions between the PI3K/AKT pathway and other signaling pathways, like JAK/STAT3 (Figure 4A,B), which may further influence bone metabolism. While this study primarily examines the PI3K/AKT signaling pathway, it is also important to investigate the role of the JAK/STAT3 signaling pathway in osteoporosis. This pathway plays a significant role in regulating bone metabolism by influencing processes such as bone resorption, formation, and remodeling, thereby maintaining the normal structure and function of bone tissue [24]. By modulating the JAK/STAT3 pathway, it is possible to inhibit bone destruction and chronic inflammation, which can lead to an improvement in osteoporosis symptoms [25]. As a sensitive molecular target, STAT3 is crucial for promoting osteoblast differentiation and enhancing the expression of osteogenic markers, ultimately contributing to both bone formation and maintenance [26]. In summary, the JAK/STAT3 and PI3K/AKT pathways are crucial in osteoporosis and have complex interactions with each other. Future research may further validate this hypothesis through the use of gene knockout models or specific inhibitors.

While this study has shown some results, it has limitations. It focused solely on the role of PI3K/AKT and its pathways in treating osteoporosis with TFRD, without considering other signaling pathways or mechanisms. Additionally, only OVX osteoporotic rats were used as models. Future research should explore other osteoporosis models (like glucocorticoid-induced or disuse osteoporosis) to assess the broader applicability of TFRD. Thus, further studies can investigate the pharmacological mechanisms of TFRD and their efficacy and safety across different types of osteoporosis, aiming to provide a more comprehensive theoretical foundation and experimental support for treatment.

## 4. Materials and Methods

### 4.1. Network Pharmacology Methods

#### 4.1.1. Database and Tool Selection

This study used several authoritative network pharmacology databases and tools, including the SwissTargetPrediction database [27] for identifying potential targets of active ingredients in TFRD, DISGENET [28], GENECARDS [29], OMIM [30], CTD [31], and TTD [32] databases for disease targets related to OP, the STRING database for constructing protein–protein interaction (PPI) networks among potential target proteins of TFRD in treating OP, and Cytoscape software for data integration and visualization.

#### 4.1.2. Targets Prediction

To predict active component targets, the main components of TFRD are input into the SwissTargetPrediction database to identify potential targets. Disease target filtering and validation involve retrieving OP-related targets from DISGENET, GENECARDS, OMIM, CTD, and TTD databases using the keyword “osteoporosis”, while removing those not directly related to drug action. A Venn diagram is created with R 4.3.1 software to identify common action targets of TFRD in treating OP.

#### 4.1.3. PPI Network Construction

Using the key targets for OP treatment from the TFRD, a PPI network was created with the STRING database and analyzed topologically. The network was visualized using Cytoscape 3.7.2.

The selected targets were entered into the STRING database, and a minimum interaction score of 0.900 was set to build the PPI network. Cytoscape 3.7.2 plugins like NetworkAnalyzer (4.5.0) were used to perform topological analysis on the PPI network and identify key nodes, such as hub nodes and modules.

#### 4.1.4. “TFRD Component-Targets” Network

The “TFRD component-Targets” network was created using Cytoscape 3.7.2 for visualization and analysis. The organized TFRD components and OP targets data were imported into Cytoscape, where they were defined as nodes, with their interactions represented as edges. Cytoscape analyzed the component–target network to reveal interaction relationships between each TFRD component and target through structural analysis. Additionally, core targets or pharmacodynamic components of TFRD were identified using methods like network clustering analysis or module mining.

#### 4.1.5. Pathway Enrichment Analysis

Pathway enrichment analysis was performed on key nodes in the PPI network using KEGG [33] and GO [34] databases to identify potential biological pathways affected by TFRD. Differentially expressed genes were annotated with the GO database, and significant GO terms were selected based on these annotations. Enrichment significance for each GO term was calculated using methods like Fisher’s exact test or chi-square test.

#### 4.1.6. “TFRD Component-Target-Pathway” Network

The PPI network and pathway enrichment analyses revealed the potential mechanisms of TFRD in treating OP. To visually represent the targets and interactions of TFRD, Cytoscape software was used.

#### 4.1.7. Grouping and Modeling of Experimental Animals

Seventy-two female SD rats, aged 10 weeks and weighing 254 ± 22 g, were purchased from Beijing Vital River Laboratory Animal Technology Co., Ltd. (Beijing, China). The experimental animal production license number is SCXK (Beijing, China) 2016-0011. This study was approved by the Experimental Animal Ethics Committee of the Institute of Basic Theory of Traditional Chinese Medicine at the China Academy of Chinese Medical Sciences, adhering to all ethical requirements for medical experiments (license number SYXK (Beijing, China) 2021-0017). All rats had free access to food and water during the experiment in a controlled environment with constant temperature and humidity, following a 12 h light/dark cycle.

After 72 rats were acclimated for 2 weeks, they were randomly divided into three groups: a control group (CON, 16 rats), a sham-operated group (SOG, 16 rats), and a model group (40 rats). The model group underwent bilateral ovariectomy (OVX) to create an osteoporosis rat model, while the SOG had only fat tissue removed. After surgery, all rats were cared for and monitored during recovery. Twelve weeks post-surgery, 8 rats from each of the three groups were anesthetized and sacrificed for sample collection. The remaining model group was split into four subgroups of 8: OVX, Sim, low-dose TFRD, and high-dose TFRD groups. The control and SOG received pure water orally; the Sim received 10 mL/kg/day of a 0.036 mg/mL estradiol valerate solution; while the TFRD groups received TFRD solutions at doses of 3.5 mg/mL and 7 mg/mL, respectively. All treatments lasted for 12 weeks to assess their effects on OP improvement. After this period, all remaining rats were anesthetized again for sample collection. Their mandibles underwent Micro-CT analysis to measure bone density and histomorphometric parameters; paraffin sections were prepared for Western blot analysis to detect protein expression in the mandibles.

#### 4.1.8. Drugs and Main Reagents

TFRD are the main component in Qianggu Capsules, produced by Beijing Qihuang Pharmaceutical Co., Ltd. (Beijing, China) (National Drug Approval No. Z20030007). Each capsule contains 0.25 g and has a batch number of 201102. The equivalent doses for rats were calculated based on the body surface area ratio between humans and rats, resulting in dosages of 35 mg/kg/day (low dose) and 70 mg/kg/day (high dose).

Estradiol valerate tablets (Climen) were obtained from the Guangzhou Branch of Bayer HealthCare Pharmaceuticals Inc. (Guangzhou, China) (National Drug Approval No. J20171038), batch number 602A, with a dosage of 1 mg per tablet. The equivalent dose for rats was calculated based on the body surface area ratio between humans and rats, resulting in a dosage of 0.36 mg/kg/day for each rat.

Antibodies like PI3K, AKT, and p-AKT were purchased from Servicebio Technology (Wuhan, China) for Western blot experiments.

#### 4.1.9. Detection Instruments for Bone Mineral Density (BMD) and Bone Morphological Indicators

Mandibular bone mineral density measurement conditions and analysis methods: Micro-computed tomography (Micro-CT, Bruker Company, Ettlingen, Germany) was employed to scan the mandibles of rats in each group. The scanning parameters were configured as follows: current at 200 µA, voltage at 70 kV, resolution at 10.2 µm, and a 0.5 mm Al filter was applied. Image reconstruction was performed using NRecon (version 1.7.4.2) software (Ettlingen, Germany). A region of cancellous bone (volume 1.0 mm × 0.8 mm × 1.5 mm) located between the roots of the first molar was defined as the region of interest (ROI) for three-dimensional reconstruction. Subsequently, DataViewer (v1.0.1.160122), CTan (v1.10), and CTvox (v2.1) (Ettlingen, Germany) software were utilized to analyze the region, generate 3D images, and calculate bone mineral density (BMD) values based on the built-in parameters of the software. The key detection indicators included the following: bone mineral density (BMD) (g/cm^3^): cancellous bone density within the ROI; bone volume (BV) (mm^3^): total bone volume in the ROI; bone volume fraction (BV/TV) (%): percentage of trabecular volume to total sample volume; trabecular thickness (Tb.Th) (mm): average thickness of trabeculae; trabecular separation (Tb.Sp) (mm): average width of medullary cavities between trabeculae; and trabecular number (Tb.N) (1/mm): number of trabeculae per millimeter.

#### 4.1.10. H&E Staining Histological Analysis

Rat mandibles were used to create paraffin sections, which were stained with H&E to observe pathological changes in bone tissue and collagen fiber content. The stained sections were examined under a microscope, and the morphology, quantity, and arrangement of trabeculae were recorded and analyzed.

#### 4.1.11. Western Blot

Rat mandibular bone tissues were collected, and proteins were extracted for Western blot analysis. The protein samples underwent SDS-PAGE electrophoresis, were separated, and transferred to PVDF membranes. The expression levels of p-PI3K, PI3K, AKT, and p-AKT proteins were detected using specific antibodies. Results were recorded and analyzed with a gel imaging system.

#### 4.1.12. Molecular Docking and Validation

Using molecular docking software like AutoDock VINA (1.5.7) (Scripps, La Jolla, CA, USA) [35] and PyMOL (v3.1.3.1) (Schrödinger, New York, NY, USA), we docked the key active components of TFRD with proteins in the PI3K/AKT pathway to simulate their binding ability and affinity. The results were validated by comparing them with literature reports and existing experimental data to confirm the interactions.

#### 4.1.13. Statistical Analysis

Data analysis was performed using SPSS 23 software (IBM, Armonk, NY, USA). Measurement data are presented as mean ± standard deviation (mean ± SD), with t-tests or ANOVA used for group comparisons. A *p* value < 0.05 indicates statistical significance.

## 5. Conclusions

This study systematically investigates the mechanism of action of TFRD in treating OVX osteoporotic rats, focusing on the PI3K/AKT signaling pathway through a combination of network pharmacology and in vivo experiments. These findings offer novel insights and methodologies for osteoporosis treatment, as well as providing a theoretical foundation and experimental evidence for the clinical application of TFRD. However, to fully elucidate the mechanism underlying the effects of these flavonoids and their potential applications in osteoporosis therapy, further research is warranted.

## Figures and Tables

**Figure 1 ijms-26-03650-f001:**
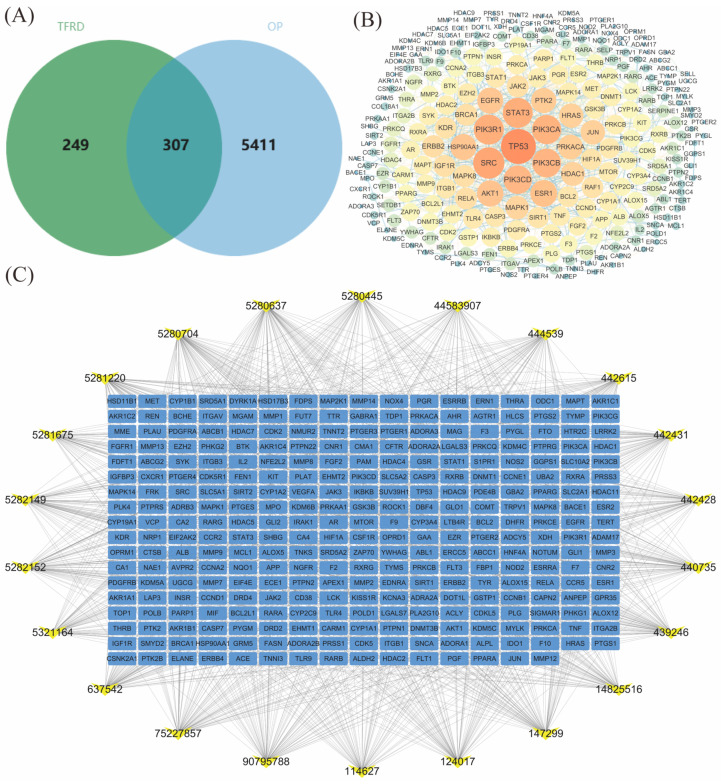
(**A**) Venn diagram identifying 307 common targets for TFRD in treating OP; (**B**) PPI networks of 307 common targets; (**C**) “TFRD Component-Targets Network” (The yellow labels in the figure represent the active ingredients, with the numbers being the PubChem CIDs, and the blue labels representing the disease targets).

**Figure 2 ijms-26-03650-f002:**
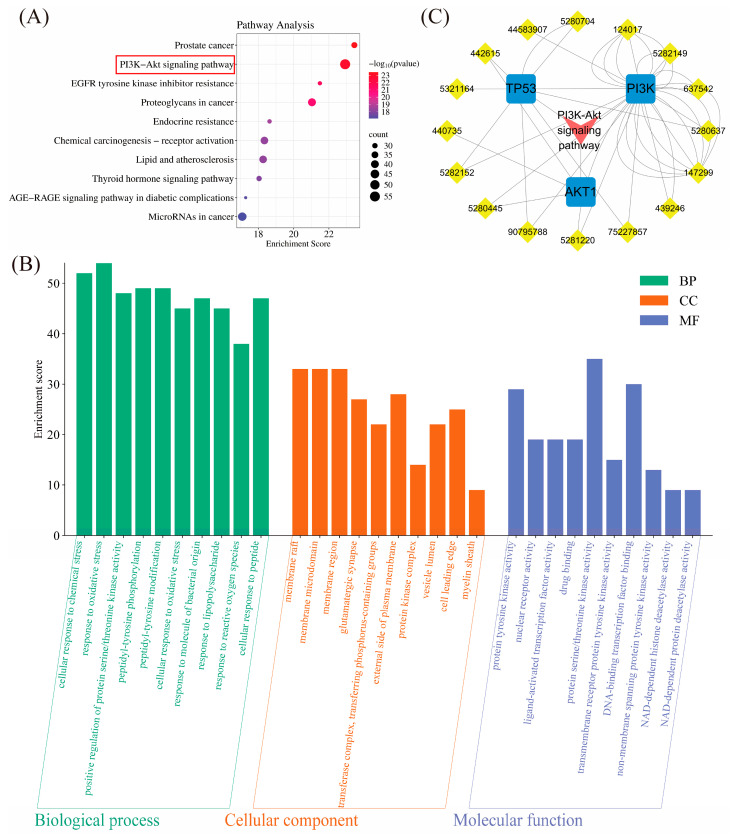
(**A**) The top 10 pathways in the KEGG pathway enrichment analysis results; (**B**) GO results of three ontologies (BP,CC,MF); (**C**) “TFRD Component-Targets-Pathway” Network (Red labels represent pathways, yellow labels represent active ingredients, and their numerical codes are PubChem CIDs, and blue labels represent targets).

**Figure 3 ijms-26-03650-f003:**
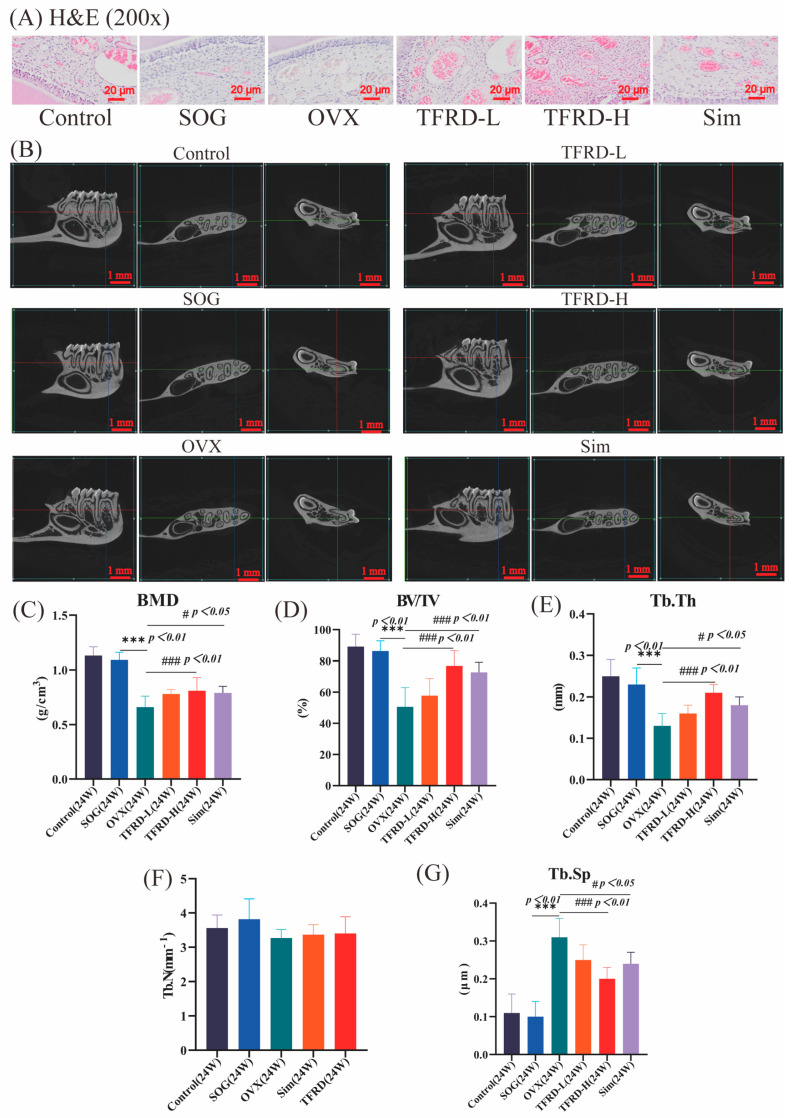
(**A**) H&E-stained images of rat mandibles (200×); (**B**) Representative Micro-CT images of the mandible of rats; (**C**–**G**) The analysis and quantification results of Micro-CT images of rat mandibles, including BMD, BV/TV, Tb.Th, Tb.N, and Tb.Sp. (The symbol “***” denotes a significance level of *p* < 0.01 when compared to the SOG group, while “#” indicates a significance level of *p* < 0.05 in comparison with the OVX group, and “###” signifies a significance level of *p* < 0.01 relative to the OVX group, *n* = 8.); Specific quantitative data are as follows: (**C**) BMD: Control—1.13 ± 0.08; SOG—1.09 ± 0.07; OVX—0.66 ± 0.1; TFRD-L—0.78 ± 0.04; TFRD-H—0.81 ± 0.12; Sim—0.79 ± 0.06; (**D**) BV/TV: Control—89.09 ± 7.96; SOG—86.26 ± 6.63; OVX—50.64 ± 12.31; TFRD-L—57.78 ± 10.99; TFRD-H—76.91 ± 9.66; Sim—72.72 ± 6.49; (**E**) Tb.Th: Control—0.25 ± 0.04; SOG—0.23 ± 0.04; OVX—0.13 ± 0.03; TFRD-L—0.16 ± 0.02; TFRD-H—0.21 ± 0.02; Sim—0.18 ± 0.02; (**F**) Tb.N: Control—3.56 ± 0.38; SOG—3.82 ± 0.59; OVX—3.27 ± 0.25; TFRD-L—3.31 ± 0.36; TFRD-H—3.4 ± 0.49; Sim—3.37 ± 0.29; (**G**) Tb.Sp: Control—0.11 ± 0.05; SOG—0.1 ± 0.04; OVX—0.31 ± 0.05; TFRD-L—0.25 ± 0.04; TFRD-H—0.2 ± 0.03; Sim—0.24 ± 0.03.

**Figure 4 ijms-26-03650-f004:**
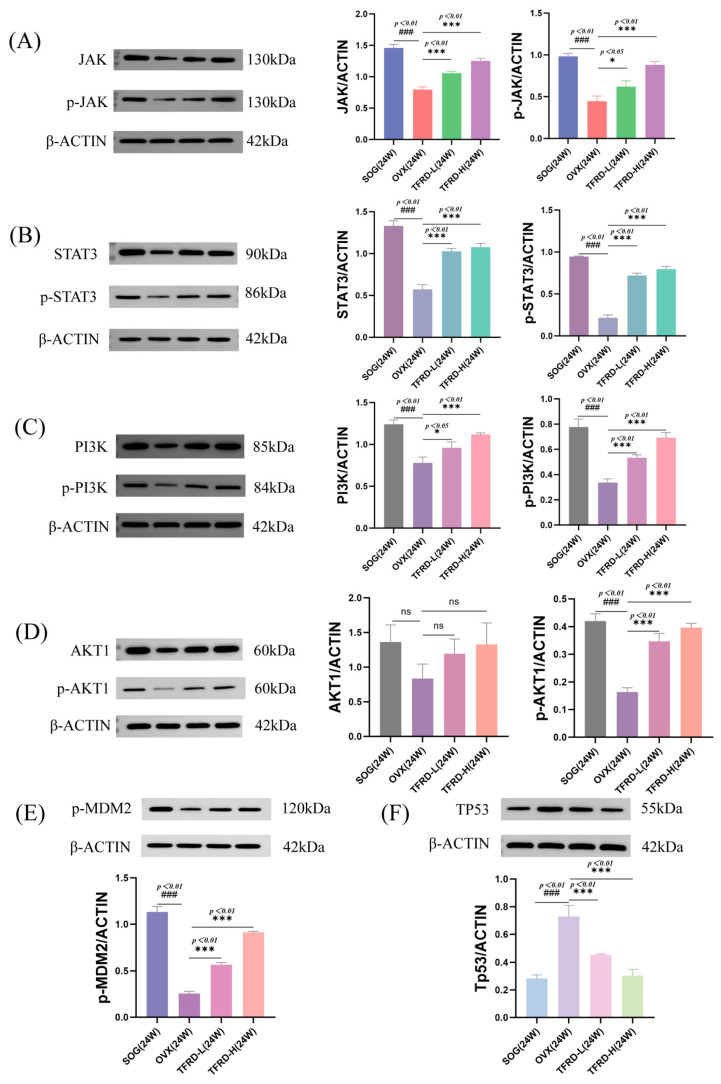
(**A**,**B**) The effect of TFRD on the expression of JAK, p-JAK, STAT3, and p-STAT3 proteins in the mandible of rats; (**C**–**F**) The effect of TFRD on the expression of PI3K, p-PI3K, AKT1, p-AKT1, p-MDM2, and TP53 proteins in the mandible of rats; (The symbol “###” denotes a significance level of *p* < 0.01 when compared to the SOG group, while “*” indicates a significance level of *p* < 0.05 in comparison with the OVX group, and “***” signifies a significance level of *p* < 0.01 relative to the OVX group; “ns” signifies no statistical significance, *n* = 3); Specific quantitative data are as follows (relative expression level): (**A**) JAK: SOG—1.46, OVX—0.7967, TFRD-L—1.057, TFRD-H—1.253; p-JAK: SOG—0.9833, OVX—0.4467, TFRD-L—0.62, TFRD-H—0.88; (**B**) STAT3: SOG—1.33, OVX—0.5733, TFRD-L—1.027, TFRD-H—1.077; p-STAT3: SOG—0.9467, OVX—0.2167, TFRD-L—0.72, TFRD-H—0.7967; (**C**) PI3K: SOG—1.24, OVX—0.78, TFRD-L—0.96, TFRD-H—1.12; p-PI3K: SOG—0.7767, OVX—0.3367, TFRD-L—0.5333,TFRD-H—0.6933; (**D**) AKT1: SOG—1.363,OVX—0.8367,TFRD-L—1.193,TFRD-H—1.327; p-AKT1: SOG—0.42, OVX—0.1633, TFRD-L—0.3467, TFRD-H—0.3967; (**E**) p-MDM2: SOG—1.133, OVX—0.2567, TFRD-L—0.5667, TFRD-H—0.9133; (**F**) TP53: SOG—0.2833, OVX—0.73, TFRD-L—0.4533, TFRD-H—0.3033.

**Figure 5 ijms-26-03650-f005:**
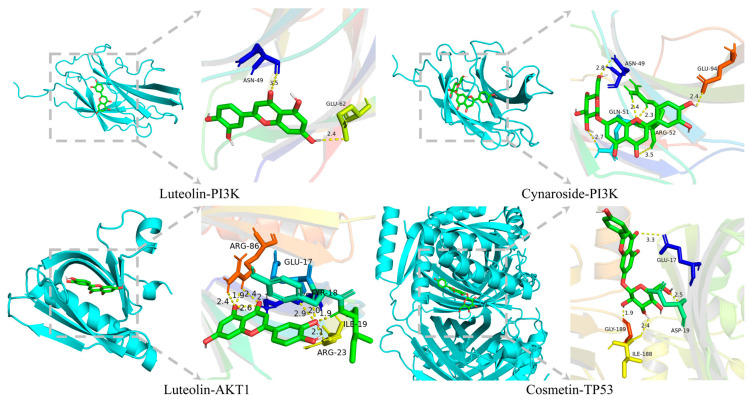
Molecular docking conformation diagrams of some active ingredients and target sites.

**Table 1 ijms-26-03650-t001:** The active components, target sites, and their binding energy results in molecular docking.

PubChem CID	Component Name	Target	Binding Energy (kcal · mol^−1^)
442,615	Lucenin-2	AKT1	−6.2
440,735	Eriodictyol	AKT1	−6
5,280,445	Luteolin	AKT1	−6.3
5,281,220	Aureusidin	PI3K	−5.8
439,246	Naringenin	PI3K	−4.8
147,299	Procyanidin B4	PI3K	−5.5
637,542	p-Coumaric acid	PI3K	−4.2
5,282,149	Trifolin	PI3K	−5.5
124,017	Procyanidin B5	PI3K	−5.4
44,583,907	5-Hydroxy-7-(beta-D-glucopyranosyloxy)chromone	PI3K	−5.3
5,321,164	Aureusidin-6-glucoside	PI3K	−4.9
5,282,152	Lonicerin	PI3K	−5.2
90,795,788	Kaempferol 3-O-Rhamnoside-7-O-Glucoside	PI3K	−3.9
75,227,857	Kaempferol 3-O-beta-D-glucopyranoside-7-O-alpha-L-arabinofuranoside	PI3K	−4.9
5,280,637	Cynaroside	PI3K	−6
5,280,445	Luteolin	PI3K	−5.9
5,282,152	Lonicerin	TP53	−10.2
90,795,788	Kaempferol 3-O-Rhamnoside-7-O-Glucoside	TP53	−10.3
75,227,857	Kaempferol 3-O-beta-D-glucopyranoside-7-O-alpha-L-arabinofuranoside	TP53	−9.4
5,280,704	Cosmetin	TP53	−10.4
442,615	Lucenin-2	TP53	−10.1
5,280,637	Cynaroside	TP53	−10.2

## Data Availability

The data presented in this study are available on request from the corresponding author. The data are not publicly available due to the confidentiality of the research project.

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
