# Peer review of "Mechanistic Integration of Network Pharmacology and In Vivo Validation: TFRD Combat Osteoporosis via PI3K/AKT Pathway Activation"

_ijms, 2025, doi:10.3390/ijms26083650_

Round 1
Reviewer 1 Report
Comments and Suggestions for Authors
Osteoporosis is a serious disease that increases the risk of bone fractures due to pathological changes in hard tissues. This problem is especially relevant for the elderly and middle-aged. Understanding the mechanisms that occur in bone tissue during osteoporosis will contribute to the search for effective treatments and drugs.
The authors of the article consider mechanistic integration of network pharmacology and in vivo validation: TFRD combat osteoporosis via PI3K/AKT Pathway Activation. The work is undoubtedly interesting, but there are a number of comments and recommendations, the solution of which will improve the quality of this manuscript and expand the circle of readers.
1. In the introduction, it is recommended to expand the information on existing methods of treating osteoporosis and the drugs used.
2. It is necessary to group the data presented in the figures differently, or divide the information into several figures, since the information on them is difficult to distinguish (in particular, histological images, signatures on histograms).
3. The manuscript contains a large number of abbreviations, but some of them (CON, SOG, GO, Sim, TFRD-L, TFRD-H, etc.) are first deciphered only at the end of the article (section 4. Materials and Methods), which complicates the understanding of the manuscript. It is recommended to introduce the abbreviation decipherment at the first mention, as well as to supplement the list of abbreviations.
4. In the second section (line 71), the authors mention their previous studies, but there are no references to these works.
5. The description of the obtained results includes only a qualitative comparison (significant increase or decrease in various indicators). The absence of a quantitative assessment complicates the analysis of the obtained results. It is still recommended to provide a quantitative assessment directly in the text of the manuscript.
Author Response
Thank you very much for taking the time to review this manuscript. Please see the attachment in the box.
Reviewer 2 Report
Comments and Suggestions for Authors
Dear Authors
The abstract does not clearly clarify the methodology used. The (abstract) conclusion is limited and without implications. You can´t say "This study supports using traditional Chinese medicine for multi-target osteoporosis treatment.", is to much!
The structure of the article is not coherent, it moves from the introduction to the results. And the methodology precedes the conclusion.
The objective is understood, but the way in which it is intended to be achieved is confusing. Despite the results being presented, using a set of figures.
Author Response

(The authors gave the same response as above.)

Reviewer 3 Report
Comments and Suggestions for Authors
It is an interesting topic.
I think it would have been better to write from the beginning what PI3K (phosphatidylinositol-3kinase) means and then just put the acronym. The same for AKT (threonine kinase). It's just my opinion to make the manuscript easier to read.
I couldn't find the study period in the text.
Lines 71-72: ,,In our previous study, we performed UPLC-MS/MS analysis on TFRD and identified 23 active components.”
I think you should have mentioned the title of the study here and possibly included references.
Lines 87-88: ,,This supports that TFRD may exert its therapeutic effects on OP through the PI3K/AKT pathway.”
Are you referring to AKT or are you thinking of one of the 3 related forms AKT1, AKT2, AKT3?
It is also known that dysregulation of the PI3K/AKT pathway is involved in a number of chronic diseases, namely cancer, cardiovascular diseases, diabetes. Have you considered this aspect as well?
Lines 136-137:,,After the administration was completed, we measured the BMD of the mandible in rats and conducted histological section observations. “
Can you specify by which method you measured the bone mineral density of the mandible in rats?
Lines 218-222: ,,In combination with the literature reports and existing experimental data, we found that Luteolin exhibited regulatory effects on the PI3K/AKT pathway in various cell models. These results further verified our molecular docking results and indicated that components such as Luteolin in TFRD might exert therapeutic effects on OVX OP through this pathway.”
My opinion is that this paragraph should be in the Discussions chapter and not in the Results. Basically, you are comparing what you have achieved in this study with the specialized literature.
Lines 250-252: ,,Our findings indicated that the BMD in the treatment group was significantly greater than that observed in the control group (P < 0.05).”
Did you find data in the literature to compare with what you obtained in your study?
It is good that you also specified the limits of the study.
Lines 354-356: ,,After 72 rats were acclimated for 2 weeks, they were randomly divided into three groups: a control group (CON, 16 rats), a sham-operated group (SOG, 16 rats), and a model group (40 rats).”
Do you also consider that a relatively equal number of rats in the three groups could influence the final results?
It is an interesting, organized paper, with a lot of up-to-date data. More studies are certainly needed.
I appreciate the work done for this study.
My comments are only intended to make the paper better. Good luck!
Author Response

(The authors gave the same response as above.)

Round 2
Reviewer 1 Report
Comments and Suggestions for Authors
Overall, the authors of the submitted manuscript have significantly improved its quality by introducing a number of clarifications. However, the quality of Figure 1 still needs to be improved, since the inscriptions on the presented images are indistinguishable even at high magnification. After this remark is corrected, the article can be published.
Author Response
Comments 1: Overall, the authors of the submitted manuscript have significantly improved its quality by introducing a number of clarifications. However, the quality of Figure 1 still needs to be improved, since the inscriptions on the presented images are indistinguishable even at high magnification. After this remark is corrected, the article can be published.
Reply 1: Thank you very much for your thorough review of our research work. Following the reviewers' suggestions, we have optimised the font size and image quality in Figure 1. Specifically, we have increased the labels' font size and enhanced the images' DPI.